# An Algorithm for the Detection of General Movements of Preterm Infants Based on the Instantaneous Heart Rate

**DOI:** 10.3390/children10010069

**Published:** 2022-12-29

**Authors:** Tomoki Maeda, Osamu Kobayashi, Eriko Eto, Masanori Inoue, Kazuhito Sekiguchi, Kenji Ihara

**Affiliations:** Department of Pediatrics, Oita University, Yufu 879-5593, Japan

**Keywords:** general movements, heart rate, accelerations, automated extraction, supine position, prone position

## Abstract

Video recording and editing of general movements (GMs) takes time. We devised an algorithm to automatically extract the period of GMs emergence to assist in the assessment of GMs. The algorithm consisted of δHR: subtracting the moving average heart rate (HR) for the past 60 s from the average instantaneous HR; and %δHR: the percentage of the instantaneous HR to the moving average HR. Ten-second sections in which δHR was positive for three consecutive sections and contained at least one section with %δHR > 105% were extracted. Extracted periods are called automated extraction sections (AESs). We evaluated the concordance rate between AESs and GMs in three periods (gestational age 24–32, 33–34, and 35–36 weeks). The records of 84 very low birth weight infants were evaluated. Approximately 90% of AESs were accompanied by GMs at any period in both the supine and prone positions. The proportion of full-course (beginning to end) GMs among GMs in the AES was 80–85% in the supine position and 90% in the prone position in all periods. We could extract a sufficient number of assessable GMs with this algorithm, which is expected to be widely used for assisting in the assessment of GMs.

## 1. Introduction

General movements (GMs) are endogenously generated spontaneous motor activities that are present from early fetal life onwards [1]. Prechtl’s method of GMs assessment (GMA) estimates the developing brain function based on video observation [2]. It is well-known that GMA is useful for the early prediction of cerebral palsy [3]. It is also reported that GMA is useful for identifying preterm infants at risk of cognitive dysfunction [4,5]. GMA does not require any special equipment other than video recording. However, the recording and editing of these videos takes time. A longer video recording is preferable for a more accurate assessment. For preterm infants, one hour of video is recommended to collect a sufficient number of GMs to make a three to five-minute video clip for the assessment that includes multiple sufficiently long examples of GMs for assessment. It is important that the video clip includes the beginning and end of each GMs [2].

Fetal movements are associated with fetal heart rate (FHR) acceleration. In the third trimester, virtually all large FHR accelerations are associated with fetal activity. The converse—that most observable fetal movements are associated with an increase in FHR—has also been reported [6]. Similarly, heart rate (HR) acceleration has been reported to be accompanied by body movement in neonates, and the acceleration in full term newborns has been reported to be greater than that in preterm newborns [7]. GMs are observed at 9 to 10 weeks of postmenstrual age onwards by fetal ultrasound. GMs are the most frequent and complex fetal movements. Fetal GMs and preterm GMs are essentially the same [1]. In preterm infants, HR should accelerate with GMs in any posture.

We have collected HR data at the same time as GMs video recordings in the preterm period. The primary purpose of simultaneous HR data sampling was for the analysis of heart rate variability (HRV). During the analysis process, we realized that instantaneous HR acceleration is well-associated with the emergence of GMs. Therefore, in this paper we devised a simple algorithm to automatically extract the periods with the emergence of GMs that can be determined from the instantaneous HR.

The instantaneous HR is easily obtained in intensive care and this algorithm-based method is expected to be widely used. This method is useful to assist in the assessment of long video clips of GMs in the neonatal intensive care unit (NICU).

## 2. Materials and Methods

### 2.1. Participants

Very low birth weight infants (VLBWIs) were recruited from the NICU of Oita University Hospital from January 2013 to April 2022. They were participants in a clinical study that investigated the association between GMs and long-term neurodevelopment (Institutional Review Board of Oita University; Approval number 515, 27 January 2012. Approval number 1288, 15 September 2017), and parental written informed consent was obtained for their participation in this study. Patients with chromosomal abnormalities and brain malformations were excluded from the clinical study.

### 2.2. General Movements (GMs) 

#### 2.2.1. Video Recording of GMs

GMs were recorded repeatedly during the NICU stay. Video recordings were performed every 2 to 4 weeks until the infant was removed from the incubator, at approximately 36 weeks of gestational age. We divided the recordings into three periods (24 to 32, 33 to 34, and 35 to 36 weeks gestation). When the recording was performed multiple times within a period, the total recording time was obtained by summing the recording times of multiple recordings. The AES and GMs during that time were added together to analyze the data for that period in one subject. We only investigated the preterm period GMs in this study because GMs after the term period are supposed to be assessed when the infant is awake. Long video recordings are usually not required after the term period.

A video camera was placed high above the incubator so that the whole body of the infant could be observed. VLBWIs were commonly in the prone position for a longer time in the preterm period. We started the video recording at approximately 30 min after tube feeding. At that time, most infants were in the prone position. After recording three to five GMs in the prone position, we changed the posture to the supine position when the infant was not in a quiet sleep state. The recording was continued in the supine position until we obtained at least five GMs recordings that were suitable for assessment [2,8]. GMs suitable for assessment means GMs of better possible quality. In order to extract GMs of better possible quality, we try to record GMs that contain various sequences. A characteristic of a good quality GM is that it begins gradually, involves the whole body, and ends gradually. The duration of such high-quality GMs is long. We select GMs lasting 2–3 min for assessment. Usually, 40 min to 1 h of recording in the supine position was required.

#### 2.2.2. Determination of the Emergence of GMs

In the neonatal period, the GMs are referred to as writhing movements. Typically, they are ellipsoid in form and are of small to moderate amplitude and speed. GMs are complex and last long enough to be observed properly. They involve the whole body in a variable sequence of arm, leg, neck, and trunk movements. They wax and wane in intensity, force, and speed, and they have a gradual beginning and end [8]. According to this definition, the emergence of GMs was decided by authors (T.M., E.E., and K.S.) by reviewing the video. E.E. and K.S. obtained certification through a basic course on GMA and T.M. obtained certification through advanced courses on GMA. The start and end times of each GMs were recorded, along with the detection of the GMs. Although there is no strict definition of the length of GMs, this defined GMs that were complex, and which lasted long enough to be observed properly [2]. The authors classified movements as GMs when the movements lasted at least 20 s from the start to the end because GMs involve the whole body and have a gradual beginning and end in variable sequences of movements. Movements that are shorter than 20 s are often isolated movements or reflex movements, such as startle responses. When long GMs were observed, we never extracted movements shorter than 20 s as GMs.

### 2.3. Heart Rate Recording

Electrocardiography (ECG) was performed using a bedside monitor (DASH4000; GE healthcare, Tokyo, Japan), with data transferred to a personal computer. The analog ECG wave output from the monitor was A/D-converted at a sampling rate of 1000 Hz using an A/D-conversion PC card (ADA16-32/2(CB)F, Contec Co., Osaka, Japan). The sequential average instantaneous HR every 10 s was outputted in the CSV format by the HRV analysis software program (MemCalc/Tonam; Suwa Trust, Tokyo, Japan).

### 2.4. Algorithm to Detect the Automated Extraction Section (AES)

We defined the “automated extraction section” (AES) as the section where we expected GMs to emerge based on the instantaneous HR. The algorithm used to detect the AES was as follows:(1)Record the average instantaneous HR of the current 10 s during the same period as GMs video recording.(2)Calculate δHR: subtracting the moving average value of the HR for the past 60 s from the average instantaneous HR of the current 10 s. %δHR is the percentage of the instantaneous HR to the moving average HR.(3)Sections in which δHR is positive for three consecutive sections and contains at least one section in which the %δHR is above 103% or 105% are extracted.(4)The section is expanded to before and after the extracted section in which the δHR becomes negative twice.

Figure 1 shows an example of a sheet showing the extraction procedure (Figure 1).

This sheet uses a %δHR threshold setting of 105% to extract AESs. Sections in which δHR are positive for three or more consecutive sections are indicated with an orange background and sections in which %δHR is above 105% are indicated with a yellow background. Sections where the two conditions overlap are extended before and after the extracted section until δHR becomes negative twice. The AES extracted by such procedures are indicated by a light blue dotted line box. Sections with the emergence of GMs are shown with a dark blue background.

HR is the average value of the instantaneous HR current over 10 s; MovAvHR is the moving average value of the HR for the past 60 s; δHR is calculated by subtracting the MovAvHR from the current HR; %δHR is the percentage of the HR to the MovAvHR.

Figure 2 shows an example of the temporal changes in extraction parameters and AES, along with the sections with the emergence of GMs (Figure 2).

### 2.5. Concordance Rate between AES and the Emergence of GMs

The GMs section determined by video observation was marked on a sheet divided by 10-s intervals. Even if GMs were not observed during the entire 10-s section, it was marked as a GMs section if it included a beginning or an end. We examined the degree of agreement between the AES and GMs sections entered chronologically on the same sheet (Figure 1).

The following four items were evaluated.

AES accompanied by GMs (%): percentage of AES accompanied by a GMs section among all AESs.Full-course GMs among GMs with AES (%): percentage of full-course GMs among GMs sections accompanied by an AES. Full-course GMs means a GMs section from the beginning to the end.GMs accompanied by an AES (%): percentage of GMs sections accompanied by an AES among all GMs sections.AES length/recording length (%): percentage of total AES time length in total video recording time length.

Sections with artifacts, such as care intervention or electrode detachment, were excluded from the examination. They could be recognized by reviewing the video.

We investigated whether it is possible to automatically extract GMs not only in the supine position, but also in the prone position.

## 3. Results

### 3.1. Participants

Within the study period, 113 VLBWIs were cared for in our NICU. There were six deaths during the neonatal period, two cases of chromosomal abnormality, and one case of congenital muscular dystrophy; these cases were excluded from the analysis. Ten cases were excluded because the parents did not give their consent for inclusion in the study. In addition, 10 cases were excluded from this study, as simultaneous HR data were not recorded until 36 weeks (gestational age). Eventually, 84 subjects were enrolled in this study. Their median gestational age was 28 weeks and 5 days (range: 23 weeks and 0 days to 34 weeks and 0 days). Their median birth weight was 1007 g (range: 406 g to 1480 g).

### 3.2. Simultaneous HR Recording with Video Recording of GMs

Simultaneous HR recordings with GMs video recordings in the supine position were collected at 29 to 32 weeks, 33 to 34 weeks, and 35 to 36 weeks in 38, 70, and 69 cases, respectively. Simultaneous HR recordings with GM video recordings in the prone position were collected at 29 to 32 weeks, 33 to 34 weeks, and 35 to 36 weeks in 18, 61, and 56 cases, respectively. The average video recording time at 29 to 32 weeks, 33 to 34 weeks, and 35 to 36 weeks was 89.9, 65.9, and 64.7 min, respectively, in the supine position, and 40.9, 35.0, and 37.0 min in the prone position (Table 1 and Table 2).

### 3.3. Concordance Rate between AES and the Emergence of GMs in the Supine Position

Table 1 presents the results of an examination of the rate of concordance between AES and the emergence of GMs in the supine position (Table 1).

The percentage of AESs accompanied by GMs was 76–83% with a %δHR threshold of 103% and approximately 90% with a %δHR threshold of 105% at any gestational age. Full-course GMs in the AES were detected more than 10 times at any gestational age. Among GMs accompanied by an AES, the proportion of full-course GM was approximately 80–85% at any gestational age. The percentage of GMs accompanied by an AES among all GMs differed by gestational age. With a %δHR threshold of 103%, the percentage was 77.4% at 24–32 weeks of gestational age, 84.2% at 33–34 weeks of gestational age, and 84.9% at 35–36 weeks of gestational age. With a %δHR threshold of 105%, the percentage was 58.4% at 24–32 weeks of gestational age, 75.7% at 33–34 weeks of gestational age, and 80.6% at 35–36 weeks of gestational age. The percentage of the total AES time length in the total video recording time length was approximately 50% with a %δHR threshold of 103% and approximately 40% with a %δHR threshold of 105% at any gestational age.

### 3.4. Concordance Rate between AESs and the Emergence of GMs in the Prone Position

Table 2 shows the results of an examination of the concordance rate between AESs and the emergence of GMs in the prone position (Table 2).

The percentage of AESs accompanied by GMs was 75–80% with a %δHR threshold of 103% and approximately 90% with a %δHR threshold of 105% at any gestational age. Full-course GMs in the AES were detected over five times at any gestational age. Among GMs accompanied by an AES, the proportion of full-course GM was approximately 90% at any gestational age. The percentage of GMs accompanied by an AES among all GMs was approximately 80% with a %δHR threshold of 103% at any gestational age. With a %δHR threshold of 105%, the percentage was 56.9% at 24–32 weeks of gestational age, 67.4% at 33–34 weeks of gestational age, and 74.5% at 35–36 weeks of gestational age. The percentage of the total AES time in the total video recording time was 40–45% with a %δHR threshold of 103% and 25–35% with a %δHR threshold of 105%.

## 4. Discussion

In the present study, we devised a simple algorithm to automatically extract the periods with the emergence of GMs that could be determined from the instantaneous HR. This algorithm method showed a high detection rate for GMs and extracted a sufficient number of GMs for assessment.

In recent years, various technology-assisted methods for the quantification of GMs have been reported [9,10]. These methods can be applied to videos clips in which GMs have already been extracted; however, they do not detect the emergence of GMs [11,12]. When aiming for a technology-assisted automatic GMs assessment, it is convenient to perform it after extracting the part that is to be assessed in detail. Even when GMs are qualitatively assessed by the classical Prechtl’s method [2], it is convenient to perform the assessment after extracting the periods with the emergence of GMs from a long video recording.

GMs involve the whole body in a variable sequence of the arm, leg, neck, and trunk movements. They wax and wane in intensity, force, and speed. The characteristics of these movements led to an increase in HR. Other movements, such as isolated hand and leg movements were short in duration and were associated with very small changes in HR. Therefore, movements shorter than GMs were rarely extracted as AES when applying this algorithm. When the infant was continuously fussing, the HR continued to be high, so there were few changes in the HR associated with movement, making it difficult to apply this algorithm. However, this point does not matter when practicing GMA, because, when the infant is agitated, it is not suitable for GMA [2].

### 4.1. The Adequate %δHR Threshold Setting

We attempted to extract AESs using two %δHR threshold settings of 103% and 105%. Using a threshold of 103%, the AES is extracted from a smaller HR acceleration in comparison to the threshold of 105%; thus, the AES was more frequently detected with a threshold of 103%. However, at this threshold, the GMs were often short with few sequences. For the purpose of automatically extracting sections with the emergence of GMs and assisting in the qualitative assessment of GMs, it is important that the AES intervals contain full-course GMs. AES were more frequently accompanied by GMs with the higher threshold of 105% at any gestational age in both the supine and prone positions. The proportion of full-course GMs were similar in the AES of 103% threshold and AES of 105% threshold at any gestational age in both the supine and prone positions. The percentage of GMs accompanied by an AES among all GMs was low, especially at 24 to 32 weeks of gestational age. However, a sufficient number of full-course GMs were detected in the AES for assessment. Moreover, the percentage of total AES time length in the total video recording time length was shorter in the AES with a threshold of 105%. From these results, we propose that a %δHR threshold of 105% be used as the default setting for AES. Since this algorithm uses the increase in instantaneous HR relative to the moving average HR as an extraction condition, it may be difficult to extract the AES when the subject moves continuously or when the basic heart rate is high. If the 105% setting does not extract a sufficient number of GMs for assessment, it might be effective to change the %δHR threshold to 103%.

### 4.2. The Effect of Advancing Gestational Age on the AES

In each evaluated item trajectory in the preterm period, GMs accompanied by an AES increased with increasing gestational age in both the supine and prone positions. It is reported that fetal motor activity becomes increasingly associated with transient accelerations of FHR as gestation advances [6]. Preterm GMs are essentially the same as fetal gross movements [1]. This result indicated that the association between GMs and HR acceleration increases as the gestational age advances in preterm infants in all positions, like fetuses.

### 4.3. The Comparison of AES in the Supine and Prone Positions

The percentage of AESs accompanied by GMs was similar in the prone and supine positions; however, the percentage of full-course GMs among AES accompanied by GMs was higher in the prone position. In addition, the proportion of AES in the total recording time was low in the prone position. This would reflect fewer spontaneous body movements during sleep and a longer interval between successive arousals in the prone position in comparison to the supine position [13,14]. In the present study, the duration of GMs was short, and the HR between GMs varied less in the prone position.

### 4.4. Future Perspectives Related to AES

As future perspectives of the AES for applying and developing the results of this study, we are considering incorporating it for the assessment of GMs and using it for an indicator of neonatal well-being.

### 4.5. The Relationship between the Traditional GMA and Characteristics of AES

Since long video recordings are required to assess GMs in the preterm period, this study only investigated the preterm period. Furthermore, we did not investigate the relationship between the results of the qualitative assessment of traditional GMs (poor repertoire, chaotic, cramped synchronized, and normal) and the amount and characteristics of AESs. This is an issue for future study. If the amount and characteristics of AESs themselves have implications other than identifying the emergence of GMs, it would be worth investigating the relationship between AESs and GMs quality in the term and post-term periods.

### 4.6. The Application to Quantitative Evaluation of GMs

The assessment of GMs is an excellent method characterized by being less invasive to the subject. As the HR is routinely monitored in the NICU, the simultaneous GMs recording and HR data collection does not add invasiveness. By applying this algorithm for the detection of the emergence of GMs based on the HR, it is possible—to some extent—to quantitatively assess GMs. In the NICU, infants are often cared for in the prone position; thus, we also performed evaluations in the prone position. Even in the prone position, we were able to extract gross movements involving the whole body that showed a high probability of corresponding to GMs. The assessment of GMs in the prone position is another future prospect. We are considering an assessment method in the prone position that includes both qualitative and quantitative indicators of GMs, based on the current study.

It might be possible to estimate the approximate amount of GMs of an infant in the NICU from the HR change without a video recording. The quantity and quality of fetal movements are an indicator of fetal well-being. It has been reported that various fetal pathological conditions are associated with a decreased quantity and worsened quality of fetal movements [1,15]. Qualitative assessment is important for the assessment of GMs by the method of Prechtl, and quantitative assessment is not emphasized [2]. However, if the quantity of GMs can be estimated from the HR, it will be possible to recognize the long-term continuous quantitative trajectories of GMs from birth and it might be used as an index to assess the current state of the brain function in a premature infant.

### 4.7. The Versatility of This Method

This algorithm can be executed using a spreadsheet software program. This is an extremely advantageous point of this method for widespread practice. We obtained an ECG for the HRV analysis at a sampling rate of 1000 Hz. However, the average HR of every 10 s used in this algorithm does not require such a high sampling rate; it is sufficient to record the instantaneous HR displayed on the bedside monitor every 10 s. The system requires the incorporation of a video camera function to record GMs with a bedside monitor. This would seem to be easily realized because neither is a special device. One limitation of this algorithm is that the exclusion of artifacts is necessary. The reason why the heart rate changes were difficult to distinguish was due to artifacts, such as care interventions or due to GMs. This was determined manually by the observation of the video. If applied without a video recording, it will be necessary to implement a system, such as manually entering a mark at the time of care intervention.

## 5. Conclusions

The algorithm to automatically extract GMs from the instantaneous HR extracted a sufficient number of GMs for assessment. This method is useful for assisting in the assessment of long video clips of GMs. The instantaneous HR is easily obtained in the intensive care setting; thus, this algorithm method is expected to be widely used.

## Figures and Tables

**Figure 1 children-10-00069-f001:**
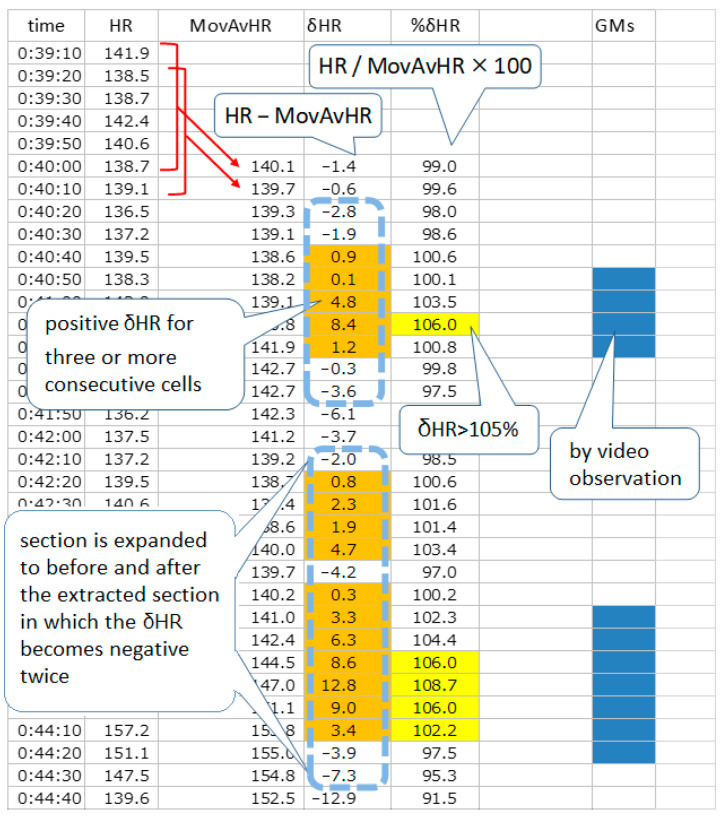
Example spreadsheet for AES extraction.

**Figure 2 children-10-00069-f002:**
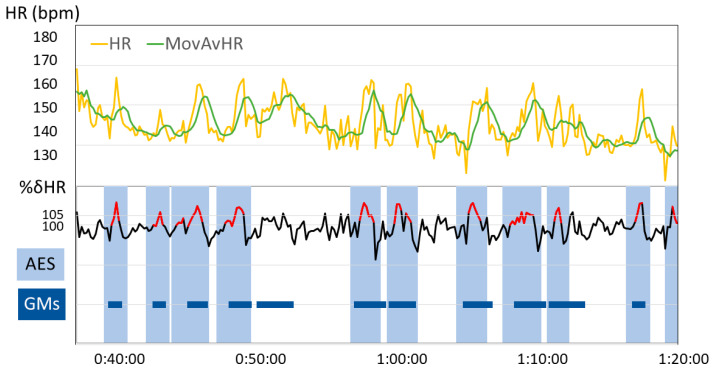
Upper row: changes over time in instantaneous heart rate (yellow line) and 60-s moving average (green line). Bottom row: change over time in %δHR (black line). The lines with positive %δHR values for three consecutive sections and at least one %δHR above 105% are highlighted in red. The AES is indicated by a light blue rectangle. The AES was extended before and after the extracted %δHR section (%δHR red line) until δHR became negative twice. The sections with the emergence of GMs are indicated with a dark blue line.

**Table 1 children-10-00069-t001:** Automatic extraction of GMs using heart rate in the supine position.

Record Gestational Age (Weeks)	24–32		33–34		35–36	
*n*	38		70		69	
Mean Record Length (Minutes)	89.9		65.9		64.7	
Extraction threshold %δHR	103	105	103	105	103	105
AES accompanied by GMs (%)	82.7	92.0	78.1	87.3	76.1	87.0
Mean number of full-course GMs in AES	17.5	13.9	11.4	10.1	12.1	11.4
Full-course GMs among GMs with AES (%)	83.1	83.5	80.7	79.6	83.6	83.0
GMs accompanied by AES (%)	77.4	58.4	84.2	75.7	84.9	80.6
AES length/recording length (%)	52.0	38.2	50.2	40.2	50.9	42.6

%δHR, the percentage of the instantaneous HR to the moving average HR; AES, automated extraction section; GMs, general movements.

**Table 2 children-10-00069-t002:** Automatic extraction of GMs using the heart rate in the prone position.

Record Gestational Age (Weeks)	24–32		33–34		35–36	
*n*	18		61		56	
Mean Record Length (Minutes)	40.9		35.0		37.0	
Extraction threshold %δHR	103	105	103	105	103	105
AES accompanied by GMs (%)	78.2	91.7	79.7	91.1	75.9	89.5
Mean number of full-course GMs in AES	7.3	5.6	6.4	5.2	6.5	5.8
Full-course GMs among GMs with AES (%)	91.0	93.5	91.8	92.4	89.5	90.8
GMs accompanied by AES (%)	78.9	56.9	82.3	67.4	83.9	74.5
AES length/recording length (%)	40.7	26.8	43.7	31.3	43.5	34.3

%δHR, the percentage of the instantaneous HR to the moving average HR; AES, automated extraction section; GMs, general movements.

## Data Availability

The datasets used in the present study are available from the corresponding author upon reasonable request.

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
