# Peer review of "An Algorithm for the Detection of General Movements of Preterm Infants Based on the Instantaneous Heart Rate"

_children, 2022, doi:10.3390/children10010069_

Round 1

Reviewer 1 Report

Summary: This article, as the title states, presents an algorithm for detection of general movements (GM) in pre-term infants based on infants’ instantaneous heart rate. GM are specific patterns of movement displayed by infants in the perinatal period that are associated with typical neurological development or impairment. GM are identified by trained observers from video and their isolation for evaluation is a laborious process. Given that heart rate is typically monitored and recorded in pre-term infant intensive care settings (i.e. in the NICU), if incidence of GM can be localized on video based on changes in HR, the process of assessing at-risk infants’ neurological status can be made more efficient. Authors describe their HR-GM mapping algorithm and its validation against the recorded GM and HR data of 84 pre-term infants.

Dear Authors:

Thank you for the opportunity to read and comment on your interesting and potentially very impactful work. The topic provides a potentially significant advantage to the clinical process of providing care for pre-term infants. The article is overall clear and well-written. Figures and tables are helpful. Figure 2 provides particular clarity.

There are some minor additions and clarifications that might make the article even more compelling. It could be improved by providing somewhat greater elaboration on both the phenomenon of general movements (GM) and the vision for operationalizing the algorithm in actual care. How are GM distinct from other infant movement, as when the infant is agitated? Given that the GMA is a proprietary instrument, we cannot assume that the general audience of Children will be uniformly familiar with it.

What work remains to be done to transform the discovery of the concordance between change in HR and GM emergence into a process that could be routinely implemented by a clinician at bedside? Readers unfamiliar with NICU routines would benefit from the authors’ making this explicit connection.

The organization of the description of the algorithm, methods and results is somewhat confusing. The description of the algorithm precedes the description of the research that (presumably) led to its development. Was the algorithm developed a priori?  Authors write at on page 2, lines 44-48 that they “have collected HR data at the same time as GMs video recordings” and that they “have confirmed that HR acceleration is well associated with the emergence of GMs.” The wording suggests that this may be a past effort, but there is no citation, creating ambiguity as to whether it refers to the current or previous work.

The reader would be interested to read in the Discussion why authors hypothesize that the HR acceleration announces the emergence of GMs. What is similar about the various types of GMs, e.g. fidgety versus cramped movements, poor repertoire, that cause them all to manifest in association with HR acceleration? Or, is this not the case? Did authors, for instance, find one type of GM to predominate in association with HR acceleration?

Finally, I noticed several typos in the manuscript, but I presume that editorial staff will flag these for authors on the back end if the article is accepted for publication.

Author Response

There are some minor additions and clarifications that might make the article even more compelling. It could be improved by providing somewhat greater elaboration on both the phenomenon of general movements (GM) and the vision for operationalizing the algorithm in actual care. How are GM distinct from other infant movement, as when the infant is agitated? Given that the GMA is a proprietary instrument, we cannot assume that the general audience of Children will be uniformly familiar with it.

Reply

Thank you for your important questions. Based on your questions, we have updated the Discussion section, as follows (page 6, line 228-236):

“GMs involve the whole body in a variable sequence of the arm, leg, neck, and trunk movements. They wax and wane in intensity, force, and speed. The characteristics of these movements led to an increase in HR. Other movements, such as isolated hand and leg movement were short in duration and were associated with very small changes in HR. Therefore, movements shorter than GMs were rarely extracted as AES when applying this algorithm.

When the infant was continuously fussing, the HR continued to be high, so there were few changes in the HR associated with movement, making it difficult to apply this algorithm. However, this point does not matter when practicing GMA, because when the infant is agitated, it is not suitable for GMA.”

What work remains to be done to transform the discovery of the concordance between change in HR and GM emergence into a process that could be routinely implemented by a clinician at bedside? Readers unfamiliar with NICU routines would benefit from the authors’ making this explicit connection.

Reply

Thank you for your important suggestions. We changed the description about the required system in the discussion to be more specific “The versatility of this method” section, as follows (page 8, line 312-314):

“This can be performed without a special data collection system.” to “The system requires the incorporation of a video camera function to record GMs with a bedside monitor. This would seem to be easily realized because neither is a special device.”

The organization of the description of the algorithm, methods and results is somewhat confusing. The description of the algorithm precedes the description of the research that (presumably) led to its development. Was the algorithm developed a priori?  Authors write at on page 2, lines 44-48 that they “have collected HR data at the same time as GMs video recordings” and that they “have confirmed that HR acceleration is well associated with the emergence of GMs.” The wording suggests that this may be a past effort, but there is no citation, creating ambiguity as to whether it refers to the current or previous work.

Reply

Thank you for your pertinent comments. The primary purpose of simultaneous HR recording was conducting HRV, not the extraction of GMs based on HR. In the analysis process, I found that an increase in HR could be used to detect the emergence of GMs, and built this algorithm. This finding has not been reported in neonates previously.

Therefore, I changed the description on page 2, line 45-47.

The primary purpose of simultaneous HR data sampling was for the analysis of heart rate variability (HRV). During the analysis process we realized that instantaneous HR acceleration is well-associated with the emergence of GMs.

The reader would be interested to read in the Discussion why authors hypothesize that the HR acceleration announces the emergence of GMs. What is similar about the various types of GMs, e.g. fidgety versus cramped movements, poor repertoire, that cause them all to manifest in association with HR acceleration? Or, is this not the case? Did authors, for instance, find one type of GM to predominate in association with HR acceleration?

Reply

Thank you for your pertinent comments. Unfortunately, we only perform long video recordings in the preterm period. Therefore, there are no data on the term period or the fidgety period. In the preterm period, the GMS that are extracted from long recordings are those that are suitable for assessment; however, the GMs after the term writhing period should be assessed when the infant is in awake. The Fidgety period is also to be assessed in the wakefulness.

We added a reason for investigation only in the preterm period in the methods section. (page 2, line 69-71). Then, we created a subheading, ``The relationship between the traditional GMA and characteristics of AES,'' under ``Future perspectives related to AES,'' and described the important theme of the relationship with qualitative assessment of GMs as a future perspectives. (page 7-8, line 279-286)

Finally, I noticed several typos in the manuscript, but I presume that editorial staff will flag these for authors on the back end if the article is accepted for publication.

Thank you for appreciating the importance of this paper.

Reviewer 2 Report

Overall, I believe this is a very useful manuscript. The information is interesting and potentially useful for those studying gereralize movements. 

There are a few minor sections of the manuscript that in my opinion need clarification.

1: lines 65-66: When recording was performed multiple times 65 within a period, multiple recordings were combined and analyzed. Please clarify what this means

2. Lines 73 what criteria were used to decide GMs recorded were suitable for assessment.

3. Lines 85-87: since there are no strict definition of length of GM why was 20 second chosen?

4. Line 205 what is a sufficient number of GMs for assessment?

5. Lines 258-259: While it would be nice to know the amount of GMs an infant demonstrates. Please clarify, I thought the purpose of GMs is to assess the quality of movement and not quantity.

6. I am confused on two aspects.

6.a. Can GMs be assessed prone? Do you have a reference for this?

6.b. Can the methodology used in this study be used clinically. Meaning, if someone set up a camera in NICU and used this methodology, then someone could then know what times they would need to go back and assess the GMs to assess the quality of the movements?

Author Response

There are a few minor sections of the manuscript that in my opinion need clarification.

1: lines 65-66: When recording was performed multiple times 65 within a period, multiple recordings were combined and analyzed. Please clarify what this means

Reply 1

Thank you for your pertinent comments. We have rewritten it with a more detailed description. The following test was added (page 2, line 67-69):

“…the total recording time was obtained by summing the recording times of multiple recordings. The AES and GMs during that time were added together to analyze the data for that period in one subject.”

  1. Lines 73 what criteria were used to decide GMs recorded were suitable for assessment.

Reply 2

Thank you for your important questions. We have added citations [2, 8] and an explanation of the term "suitable for assessment" (page 2, line 78-83).

The explanations are as follows:

GMs suitable for assessment means GMs of better possible quality. In order to extract GMs of better possible quality, we try to record GMs that contain various sequences. A characteristic of a good quality GM is that it begins gradually, involves the whole body, and ends gradually. The duration of such high-quality GMs is long. We select GMs lasting 2-3 minutes for assessment.

  1. Lines 85-87: since there are no strict definition of length of GM why was 20 second chosen?

Reply 3

Thank you for your important questions. We added a detailed explanation regarding the reason why 20 seconds was chosen (page3, line 94-101).

Although there is no strict definition of the length of GMs, this defined GMs that were complex and which lasted long enough to be observed properly. The authors classified movements as GMs when the movements lasted at least 20 seconds from the start to the end because GMs involve the whole body and have a gradual beginning and end in variable sequences of movements. Movements shorter than 20 seconds are often isolated movement or reflex movements, such as startle responses. When long GMs were observed, we never extracted movements shorter than 20 seconds as GMs.

  1. Line 205 what is a sufficient number of GMs for assessment?

Reply 4

We have described it in the Methods section (page 2, line78). At least 5 GMs were required for assessment.

  1. Lines 258-259: While it would be nice to know the amount of GMs an infant demonstrates. Please clarify, I thought the purpose of GMs is to assess the quality of movement and not quantity.

Reply 5

Thank you for your important point. As you point out, the quality of GMs should be assessed, not the quantity. In the Discussion section, since the quantity of fetal movements equivalent to preterm GMs is an indicator of fetal well-being, quantitative assessment based on long recordings could be a new indicator of well-being in premature infants. This is one way that the results of this paper may be applied in the future.

I added the heading “Future perspectives related to AES” to make this clear (page7 line 274, page8 line 287-306 ).

  1. I am confused on two aspects.

6.a. Can GMs be assessed prone? Do you have a reference for this?

Reply 6.a.

Thank you for your important point. As you point out, GMs are supposed to be assessed in the supine position. On the other hand, preterm infants in the NICU spend more time in the prone position than in the supine position. Although no assessment method has been established, GMs also emerge in the prone position. The assessment of GMs in the prone position is another future prospect. We are considering an assessment method that includes both qualitative and quantitative indicators of GMs based on the current study.

We have already described this in the paragraph entitled, "The application to quantitative evaluation of GMs." The paragraph heading was changed to “Future perspectives related to AES“ to make it easier to understand  (page7 line 274, page8 line 292-306 ).

6.b. Can the methodology used in this study be used clinically. Meaning, if someone set up a camera in NICU and used this methodology, then someone could then know what times they would need to go back and assess the GMs to assess the quality of the movements?

Reply 6.b.

Yes, it can. That is the message I would like to convey in this paper.